# The GABA and GABA-Receptor System in Inflammation, Anti-Tumor Immune Responses, and COVID-19

**DOI:** 10.3390/biomedicines11020254

**Published:** 2023-01-18

**Authors:** Jide Tian, Daniel L. Kaufman

**Affiliations:** Department of Molecular and Medical Pharmacology, UCLA School of Medicine, University of California, Los Angeles, CA 90095-1735, USA

**Keywords:** GABA, GABA_A_-receptor, GABA_B_-receptor, type 1 diabetes, multiple sclerosis, homotaurine, cancer, COVID-19, SARS-CoV-2, long COVID

## Abstract

GABA and GABA_A_-receptors (GABA_A_-Rs) play major roles in neurodevelopment and neurotransmission in the central nervous system (CNS). There has been a growing appreciation that GABA_A_-Rs are also present on most immune cells. Studies in the fields of autoimmune disease, cancer, parasitology, and virology have observed that GABA-R ligands have anti-inflammatory actions on T cells and antigen-presenting cells (APCs), while also enhancing regulatory T cell (Treg) responses and shifting APCs toward anti-inflammatory phenotypes. These actions have enabled GABA_A_-R ligands to ameliorate autoimmune diseases, such as type 1 diabetes (T1D), multiple sclerosis (MS), and rheumatoid arthritis, as well as type 2 diabetes (T2D)-associated inflammation in preclinical models. Conversely, antagonism of GABA_A_-R activity promotes the pro-inflammatory responses of T cells and APCs, enhancing anti-tumor responses and reducing tumor burden in models of solid tumors. Lung epithelial cells also express GABA-Rs, whose activation helps maintain fluid homeostasis and promote recovery from injury. The ability of GABA_A_-R agonists to limit both excessive immune responses and lung epithelial cell injury may underlie recent findings that GABA_A_-R agonists reduce the severity of disease in mice infected with highly lethal coronaviruses (SARS-CoV-2 and MHV-1). These observations suggest that GABA_A_-R agonists may provide off-the-shelf therapies for COVID-19 caused by new SARS-CoV-2 variants, as well as novel beta-coronaviruses, which evade vaccine-induced immune responses and antiviral medications. We review these findings and further advance the notions that (1) immune cells possess GABA_A_-Rs to limit inflammation in the CNS, and (2) this natural “braking system” on inflammatory responses may be pharmacologically engaged to slow the progression of autoimmune diseases, reduce the severity of COVID-19, and perhaps limit neuroinflammation associated with long COVID.

## 1. Introduction

GABA and GABA-Rs play key roles in neurotransmission and neurodevelopment in the central nervous system (CNS) and have been the subject of intensive investigations for decades by neuroscientists. Only relatively recently have the presence and functions of GABA-Rs on immune cells become evident. There are a growing number of studies reporting that the activation of GABA-Rs on human and rodent immune cells limits their pro-inflammatory activities while also promoting Treg responses and shifting APC toward anti-inflammatory phenotypes. Thereby, GABA-Rs are a natural immune cell braking system that has been pharmacologically targeted to ameliorate inflammatory disorders in preclinical models. Here, we review GABA-R physiological functions in the CNS, GABA-R-based therapeutics for autoimmune diseases and inflammatory disorders along with its currently understood mechanisms of action, GABA’s role as an immunosuppressant within some solid tumors, and recent observations of GABA-R agonists’ potential as therapeutics for COVID-19. Finally, we further advance the hypotheses that T cells express GABA_A_-Rs to limit inflammation in the CNS and that GABA_A_-R agonists could aid in the treatment of CNS inflammatory disorders including MS and neuroinflammation associated with long-COVID.

## 2. GABA and GABA-Receptors and Their Physiological Functions in the CNS

In vertebrate neurons, GABA is synthesized through the decarboxylation of glutamate by two glutamic acid decarboxylases (GAD65 and GAD67). These two isoforms of GAD are highly homologous but have distinct kinetic properties, cofactor requirements, and subcellular locations, allowing the fine-tuning of GABA production with neuronal activity and cellular metabolism [1,2,3,4,5]. There are two types of GABA-Rs that are evolutionarily and functionally unrelated: GABA_A_-Rs and GABA_B_-Rs. GABA_A_-Rs are pentamers of 19 possible subunits that form fast-acting chloride channels, which when activated let chloride passively flow in or out of the cell. The direction of Cl^−^ flux depends on the equilibrium potential of Cl^−^ relative to the cell membrane potential, which in turn is modulated by the activity of the cell’s Cl^−^ transporters [6,7,8]. The influx/efflux of Cl^−^ through GABA_A_-Rs modulates the levels of intracellular ions and consequently many signaling pathways involved in neurodevelopment and neurotransmission [6,7,8]. Different combinations of GABA_A_-R subunits form GABA_A_-Rs with varying pharmacological sensitivities and kinetics. The GABA_B_-Rs are heterodimers that form a slow-acting G-protein-coupled receptor that modulates adenylyl cyclase activity as well as potassium and calcium channels [9,10].

GABA_A_-Rs and GABA_B_-Rs have different pharmacologically targetable sites which can modulate their sensitivity and function [6,10]. Orally consumed GABA has little or no ability to pass through the blood–brain barrier (BBB) [11,12,13,14]. Therefore, many BBB-permeable GABA_A_-Rs and GABA_B_-Rs agonists and positive allosteric modulators (PAMs), such as benzodiazepines and barbiturates, were developed and have provided first-line treatments for neurological and neuropsychiatric disorders including seizures, anxiety, schizophrenia, and insomnia [6,8]. The neurological activity of these BBB-permeable drugs is an inherent drawback to their use when seeking to target peripheral GABA-Rs. In contrast, oral GABA can modulate peripheral GABA-Rs without CNS side effects. Additionally, while the GABA_A_-R-specific agonist homotaurine is BBB-permeable, it had no adverse CNS effects when tested in a large phase III clinical trial for Alzheimer’s disease and may also be an attractive candidate for modulating GABA_A_-Rs in both the periphery and the CNS, as discussed below.

## 3. GABA-R-Based Therapies for T Cell-Mediated Autoimmune Diseases and Other Inflammatory Disorders

In 1999, GABA was reported to inhibit T cell proliferation in vitro and that this was mediated through GABA_A_-Rs, but not GABA_B_-Rs. In vivo, GABA administration modestly limited classical Th1-mediated delayed-type hypersensitivity (DTH) responses [15]. The anti-inflammatory effect of GABA was mediated at least in part through inhibiting TCR-mediated T cell cycling without affecting cell viability [16]. It was proposed that because of the relatively high concentration of intracellular Cl^−^ in T cells, the activation of T cell GABA_A_-R Cl^−^ channels leads to Cl^−^ efflux, depolarization, and limitation of Ca^2+^ entry into the cell to reduce intracellular calcium levels, thereby reducing T cell proliferation [15,16], a notion that has been supported by studies of Ca^2+^ flux [17,18,19,20].

The observations that GABA could limit T cell proliferation in vitro and DTH responses in vivo suggested that GABA administration could help abate autoimmune processes in which autoreactive Th1 and/or Th17 cells mediate tissue damage. Indeed, a growing number of studies have now shown that treatment with GABA or the GABA_A_-R agonist homotaurine can inhibit the progression of a diverse array of T cell-mediated autoimmune diseases that occur in rodents with different genetic backgrounds, including models of multiple sclerosis (MS), type 1 diabetes (T1D), rheumatoid arthritis, Sjogren’s syndrome, as well as inflammation in type 2 diabetes [16,20,21,22,23,24,25,26,27]. Mechanistic studies revealed that treatment with a GABA_A_-R agonist in these disease models inhibited the inflammatory activities of rodent and human Th17 and Th1 CD4^+^ T cells and cytotoxic CD8^+^ T cells [15,16,18,19,23,24,25,28,29] and also promoted CD4^+^ and CD8^+^ Treg responses [20,22,23,24,28] (Figure 1). These differential outcomes may be due in part to Th1 and Th17 cells having the greatest dependency on Ca^2+^ influx for their activation and effector functions, while Tregs display the least dependency on Ca^2+^ influx [30,31,32,33,34,35]. Long-term GABA treatment did not desensitize antigen-specific T cell responses to cognate antigen upon re-exposure [21], indicating that GABA treatment did not exhaust pre-existing antigen-specific memory T cells. 

While early studies of GABA-Rs in the immune system focused on T cells and autoimmune diseases, it became increasingly apparent that cells of the innate immune system also expressed GABA_A_-Rs and GABA_B_-Rs that modulated their function. As observed in the adaptive immune system, the activation of GABA_A_-Rs on macrophages, dendritic cells (DC), and NK cells of the innate immune system reduces their inflammatory activities and shifts them toward anti-inflammatory phenotypes [19,21,25,27,36,37,38,39,40,41]. Moreover, GABA_B_-R agonists were found to (1) inhibit murine DC activation and immune cell chemotaxis [42,43], (2) inhibit DC proinflammatory functions [42], (3) suppress antigen presentation activity of DCs to alleviate collagen-induced arthritis and contact dermatitis in mouse models [42,43], (4) inhibit the destruction of insulin-producing ß-cells in T1D-prone mice [44,45], and (5) attenuate TLR4-induced inflammatory signaling in human peripheral blood mononuclear cells (PBMCs) [46]. Although classical neuronal GABA_B_-Rs are heterodimers of GABA_B1_ and GABA_B2_ subunits, human immune cells express relatively high levels of the GABA_B1_ gene but little or no GABA_B2_ transcripts [45,47]. There is growing evidence that GABA_B1_ subunits have functional activities independent of GABA_B2_ subunits [48,49,50,51], but its mechanisms have not been elucidated.

A major factor driving relapses of human MS and experimental autoimmune encephalomyelitis (EAE, a model of MS) is thought to be the spreading of T cell autoreactivity to new myelin T cell determinants within the CNS [52,53,54,55]. While oral GABA administration did not affect EAE disease progression, presumably because it has little or no ability to pass through the BBB, treatment with the BBB-permeable GABA_A_-R-specific agonist homotaurine effectively induced disease remission in mice with advanced relapsing-remitting EAE [24,25]. Mechanistically, homotaurine treatment limited the spreading of T cell autoreactivities to new myelin antigen determinants, decreased autoreactive Th17 and Th1 responses in the CNS, and also enhanced CD4^+^ and CD8^+^ regulatory T cell responses [24,25]. The remission of EAE may have also involved the attenuation of the inflammatory activities of microglia, astrocytes, and inflammatory infiltrates in the CNS, which also possess GABA_A_-Rs and contribute to EAE pathogenesis [56,57,58,59,60,61]. In other studies of rats with an autism spectrum-like disorder caused by prenatal exposure to valproic acid [62], homotaurine treatment reduced the levels of IL-1ß, IL-6, and TNF-α, but increased the levels of IL-10 in the CNS, decreased neuronal loss, and ameliorated behavioral deficits [63].

In large drug screening assays, homotaurine was found to physically interfere with amyloid aggregation in vitro, making it a candidate treatment for Alzheimer’s disease. In a large phase III clinical study with Alzheimer’s patients, homotaurine (also known as Tramiprosate) treatment over 1.5 years failed to meet primary endpoints, but the treatment appeared to be very safe [64,65,66]. A follow-up MRI study of a subgroup of these patients indicated that homotaurine treatment slowed hippocampal atrophy with some evidence of a beneficial effect on cognition [67]. A recent small clinical study of individuals with mild cognitive impairment found that those who were given homotaurine for 12 months had reduced circulating levels of the proinflammatory cytokine IL-18 but increased levels of IL-10 and IL-33, and improved short-term memory performance [68]. Hence, homotaurine appears to be a promising candidate for testing in clinical trials for MS as well as other disorders in which inflammation in the CNS contributes to disease pathogenesis.

Recent metabolomic studies of murine T cell development have found that GABA also modulates murine T cell differentiation [69]. Depending on the context, intracellularly synthesized GABA can enter the TCA cycle to enhance bioenergetic capacity and the differentiation of Th17 cells, or GABA may be secreted where it acts in an autocrine manner to promote inducible Treg (iTreg) differentiation [69].

GABA also inhibits the secretion of pro-inflammatory cytokines and chemokines by human immune cells in vitro [19,70]. Consistent with those findings, in a small clinical trial (NCT02002130), treatment of newly diagnosed T1D children with GABA reduced their PBMC responses to a ß-cell autoantigen compared to PBMC from placebo-treated children over the 12-month study (Dr. Hubert Tse, University of Alabama at Birmingham, personal communication). Therefore, GABA-R agonists may be promising candidate therapeutics for human T cell-mediated autoimmune diseases.

## 4. Are T Cell GABA_A_-Rs a Braking System against T-Cell-Mediated Inflammation in the CNS? Lessons from Multiple Sclerosis Models

The presence and anti-inflammatory activities of GABA_A_-Rs on T cells raise basic questions as to (1) why T cells evolved to express receptors for this neurotransmitter, (2) what purpose do GABA_A_-Rs on T cells serve in states of health, and (3) where are T cells modulated by endogenous GABA? Given that the CNS is by far the major site of GABA production and secretion, we hypothesized that GABA_A_-Rs on T cells may be a natural braking system against inflammation in the CNS [16,24,25].

The vesicular release of GABA from presynaptic neurons in the CNS leads briefly to levels of ≈1 mM GABA within the synaptic cleft [71,72,73]. The GABA_A_-R subtypes on the postsynaptic membrane are comprised of subunits that confer a relatively low affinity for GABA. This allows the GABA to rapidly dissociate from the GABA_A_-Rs and be rapidly taken into cells again by transporters so that the postsynaptic neuron can quickly respond to the next release of GABA from presynaptic terminals. Some GABA can spill over from the synaptic cleft and GABA is also released from (1) neurons in a non-vesicular action potential-independent fashion from extrasynaptic sites and (2) glia [60,61,74], such that the extracellular GABA levels in the CNS are thought to be in the nanomolar to low micromolar range [75,76,77]. Notably, the GABA_A_-R subtypes on cells in the periphery, such as T cells, are composed of subunits that confer a high affinity for GABA (in the nanomolar range) [78,79,80,81,82,83,84]. Accordingly, the levels of extracellular GABA in the CNS should be sufficient to activate the high-affinity GABA_A_-Rs on CNS-infiltrating T cells. 

The notion that GABA in the CNS may limit the inflammatory activities of infiltrating activated T cells is supported by the observation that GABA_A_-R-deficient myelin-specific T cells mediated a more severe EAE than their GABA_A_-R-expressing counterparts [69]. This suggests that extracellular CNS GABA had some inhibitory effect on infiltrating GABA_A_-R-expressing encephalitogenic T cells; however, the degree of inhibition was insufficient to prevent the robust experimentally induced autoreactive T cell responses from mediating EAE, although it was less severe. The aforementioned findings that administering the BBB-permeable GABA_A_-R agonist homotaurine, but not GABA (BBB-impermeable), led to EAE remission [24,25] indicate that pharmacological augmentation of the endogenous GABA_A_-R braking system is needed to control this acute neuroinflammatory disease. EAE remission may have also involved homotaurine-mediated diminution of the inflammatory activities of microglia, astrocytes, and other inflammatory infiltrates in the CNS, all of which express GABA_A_-Rs and contribute to EAE pathogenesis [56,57,58,59,60,61]. Together, these observations suggest that endogenous GABA acts as a natural braking system on inflammation in the CNS and that this can be pharmacologically augmented.

## 5. GABA Inhibits Tumor-Infiltrating Immune Responses and Modulates Tumor Cell Proliferation

Recent studies have made the surprising finding that B cells (but not T cells) express GAD67 and secrete GABA [19]. The B cell-secreted GABA was found to inhibit antitumor responses by suppressing the infiltration and activity of cytotoxic T cells and macrophages. Tumors within B cell-deficient mice had more tumor-infiltrating CD8^+^ T cells with enhanced cytotoxic and inflammatory marker expression. Moreover, treatment of wildtype mice with a GABA_A_-R-specific antagonist significantly increased the cytotoxic activity of tumor-infiltrating CD8^+^ T cells and inflammatory marker expression in tumor-associated macrophages. Tumors implanted into mice with GAD67-deficient B cells had reduced growth, and the tumor-infiltrating CD8^+^ T cells had greater cytotoxicity and stronger pro-inflammatory properties than did CD8^+^ T cells isolated from tumors in wildtype animals [19]. These findings suggest that pharmacological inhibition of T cell GABA_A_-Rs, or the adoptive transfer of GABA_A_-R-deficient tumor-reactive T cells or APCs may enhance the efficacy of anti-cancer treatments [19,85,86]. These observations also lend support to the notion that extracellular GABA in the CNS may contribute to the immunosuppressive environment.

The activation of GABA-Rs and GABA_A_-R PAMs has also been found to modulate the metabolism and signaling pathways of some tumor cells, resulting in the inhibition or promotion of their replication and migration. The particular outcome of GABA-R activation on tumor cell proliferation and migration has varied depending on the type of tumor and the type of GABA-R and its subunit composition (recently reviewed in [87]). Benzodiazepine use has been epidemiologically associated with increased risk for some cancers [88,89]; however, care should be taken in interpreting those findings since some benzodiazepines, such as diazepam, have high affinity for the mitochondrial translocator protein (TSPO), previously known as the “peripheral benzodiazepine-binding receptor,” which is widely expressed and modulates many cellular processes [90].

A recent study found that some tumor cells express GAD67 and secrete GABA [85]. This tumor-secreted GABA can (1) activate GABA_B_-Rs on tumor cells, which facilitates GSK-3β inactivation and leads to enhanced β-catenin signaling and tumor growth, and (2) suppresses tumor-infiltrating CD8^+^ T cells [85]. The above studies demonstrate that the GABA/GABA-R system can help tumor cells escape from immune surveillance in different types of solid tumors. Accordingly, the targeting of GAD67/GAD65 or specific GABA-R subunits might be of therapeutic value for limiting the progression of these types of tumors. We also speculate that the ability of glioblastomas and neuroblastomas to produce and secrete GABA ([60,91,92,93,94,95,96,97], and our unpublished observations) could contribute to tumor cell proliferation and the immunosuppressive environment in these types of malignant tumors. 

## 6. GABA-R Agonists Modulate Airway Epithelial Cells and Pulmonary Immune Cells

Both GABA_A_-R and GABA_B_-R expression has been observed in rodent and human lung epithelia [98,99,100]. Although there are some variations in observations between studies, it appears that rodent and human airway epithelial cells express GAD67 (primarily) and GAD65, with GAD67 found in alveolar epithelial type II (ATII) but not ATI cells [98,101,102]. GABA_A_-Rs have been demonstrated on rat ATII cells electrophysiologically, which acted to increase Cl^−^ efflux [102].

GABA and GABA_A_-R PAMs reduce inflammation and improve alveolar fluid clearance and recovery of lung function in different rodent models of acute lung injury [100,103,104,105,106,107,108,109]. Likewise, GABA_A_-R PAMs reduce pulmonary inflammation and improve clinical outcomes in ventilated human patients [110,111,112]. Of potential relevance to COVID-19 treatment, GABA decreases the secretion of inflammatory factors from cultured human bronchial epithelial cells [104] and GABA_A_-R PAMs can reduce the numbers of macrophages and the levels of inflammatory cytokines in bronchoalveolar lavage fluid, as well as rodent and human macrophage inflammatory responses [36,37,108,113,114,115,116]. GABA_A_-R ligands also inhibit (1) the activation of human neutrophils and neutrophil extracellular trap (NET) formation [117,118,119] and (2) platelet aggregation [120], both of which contribute to pulmonary thrombosis.

## 7. GABA_A_-Rs Agonists Are Effective Therapeutics in Preclinical Models of COVID-19 

The ability of GABA_A_-R agonists and potentiators to limit inflammation and acute lung injury in vivo suggests that they have the potential to reduce some of the major contributors to the development of severe COVID-19. On the other hand, their anti-inflammatory activities may limit key antiviral responses and exacerbate the severity of the infection.

Murine Hepatitis Virus Strain 1 (MHV-1) studies. Like severe acute respiratory syndrome coronavirus 2 (SARS-CoV-2), MHV-1 is a beta coronavirus and was used as a model of SARS-CoV infection [121,122,123,124]. Wildtype A/J mice infected with MHV-1 develop severe pneumonitis with high mortality [121,122,123,124]. We observed that oral GABA treatment just after MHV-1 inoculation, or 3 days later when clinical symptoms had begun to appear, protected 80–90% of virus-infected mice from developing severe disease [125]. Additionally, GABA treatment modestly reduced viral load in the lungs and greatly reduced lung inflammation in mice [125]. Follow-up studies using the GABA_A_-R-specific agonist homotaurine, or the GABA_B_-R-specific agonist baclofen, revealed that homotaurine, but not baclofen, was as effective as GABA in preventing severe disease and limiting pneumonitis, indicating that GABA_A_-Rs primarily mediated GABA’s therapeutic benefits in MHV-1 infected mice [125].

SARS-CoV-2 studies. Because MHV-1 and SARS-CoV-2 bind to different cellular receptors (CEACAM1 and ACE2, respectively) and the immune response to virus infections varies greatly depending on the virus strain [126], we sought to more stringently test the therapeutic potential of GABA as a therapy for COVID-19. We therefore studied transgenic K18-hACE2 mice which express human ACE2. Following infection with SARS-CoV-2, K18-hACE2 mice develop severe pneumonitis, providing a widely used acute and lethal model of COVID-19 [127,128,129,130,131,132]. We found that oral GABA administration beginning just after SARS-CoV-2 infection, or a few days later near the peak of lung viral load, reduced disease severity, pneumonitis, and death rates in K18-hACE2 mice [133]. Notably, GABA-treated mice had reduced viral loads in their lungs (23-fold lower) and displayed shifts in circulating cytokine and chemokine levels that are associated with better outcomes in COVID-19 patients [133]. Thus, administration of this GABA-R agonist had multiple beneficial effects that are also desirable for the treatment of COVID-19 (Figure 1). 

The ability of GABA treatment to modestly reduce viral loads in both MHV-1 and SARS-CoV-2-infected mice was unexpected. The activation of GABA_A_-Rs leads to the influx or efflux of Cl^−^ depending on the equilibrium potential of Cl^−^ relative to the cell membrane potential. Consequently, activating GABA_A_-Rs on adult neurons or islet α-cells leads to Cl^−^ influx and hyperpolarization, while activation of GABA_A_-Rs on T cells, islet ß-cells, and lung ATII cells, causes Cl^−^ efflux and depolarization [17,18,19,102,115,134,135]. Coronaviruses promote Ca^2+^ influx into host cells to enhance their replication [136,137]. Conceivably, the activation of GABA_A_-Rs on lung epithelial cells leads to Cl^−^ efflux and membrane depolarization, which limits the influx of extracellular Ca^2+^ and reduces intracellular calcium levels, making the intracellular environment less conducive to viral replication. This scenario, as well as possible GABA-R-mediated changes in virus receptor levels, surfactant production/absorption, and/or inflammatory responses and autophagy, warrants further investigations (Table 1).

**Figure 1 biomedicines-11-00254-f001:**
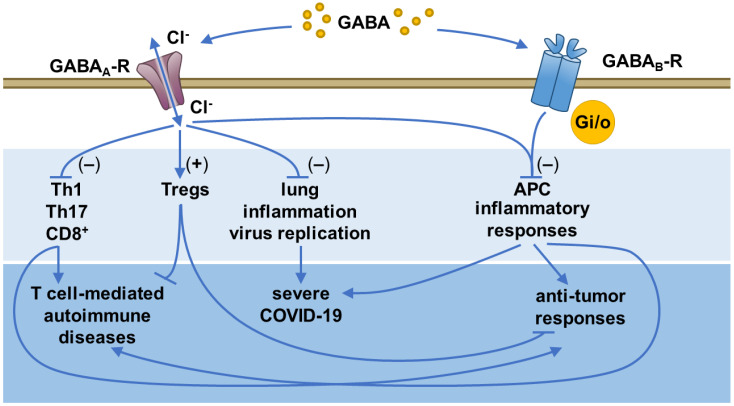
**The effects of the GABA and GABA-R system on inflammatory responses in different disease states.** GABA_A_-R agonists (1) inhibit pathogenic/effector Th1, Th17, and CD8^+^ T cell responses, reducing the production of IFNγ, IL-17, and other pro-inflammatory mediators, (2) limit APC inflammatory activities and shift them toward anti-inflammatory phenotypes, and (3) promote CD4^+^ and CD8^+^ Treg responses to enhance the production of anti-inflammatory IL-10 and other mediators. Moreover, GABA_B_-R agonists inhibit murine APC activation and proinflammatory functions. These actions limit the pathogenesis of T-cell-mediated autoimmune diseases, such as T1D, MS, rheumatoid arthritis, and Sjogren’s syndrome, as well as inflammation in type 2 diabetes, in preclinical models. Within certain types of tumors, these actions attenuate antitumor responses. Furthermore, GABA_A_-Rs agonists limit lung inflammation, virus replication, and inflammatory cytokine/chemokine responses to reduce the severity of disease and mortality in MHV-1 and SARS-CoV-2 mouse models of COVID-19. Therefore, besides the roles GABA and GABA-Rs play in neurotransmission in the CNS and in regulating pancreatic islet cell functions, they also act as brakes on inflammatory immune responses and may be a therapeutic target for the intervention of inflammatory diseases.

## 8. Could Homotaurine’s Anti-Inflammatory Effects in the Periphery and in the CNS Help Limit the Neurological Sequelae of COVID-19?

Many COVID-19 patients suffer from long-term neurological symptoms. These post-acute sequelae of COVID-19 frequently involve cognitive impairments (“brain fog”), difficulty concentrating, anxiety, and depression. Histological studies of brains obtained from COVID-19 patients have noted immune cell infiltrates and increased frequencies of glial cells with inflammatory phenotypes, which are indicative of neuroinflammatory responses [138,139,140,141,142,143,144,145]. Currently, there are no specific treatments for this condition.

We speculate that homotaurine is a good candidate for clinical trials with individuals suffering from neurological sequelae of COVID-19 based on (1) the ability of homotaurine to inhibit autoreactive T cell responses and ameliorate advanced EAE [24,25], (2) microglia and astrocytes also express GABA_A_-Rs, which act to downregulate their inflammatory activities [60], (3) GABA-R agonists protected MHV-1and SARS-CoV-2-infected mice from severe illness and reduced the levels of circulating inflammatory cytokines that might enter the CNS [125], (4) homotaurine treatment decreased proinflammatory and increased anti-inflammatory cytokine profiles in the CNS, preserved neurons, and ameliorated behavioral deficits in rats with autism spectrum-like disorder [63], and (5) Alzheimer’s patients who were given homotaurine had reduced levels of circulating inflammatory cytokines and improved short-term memory performance [68]. Thus, homotaurine has the potential to reduce dysregulated immune responses to SARS-CoV-2, reduce viral loads, and act directly in the CNS to limit the inflammatory activities of glial cells and infiltrating immune cells. As noted above, homotaurine was found to be very safe in a large long-term phase III clinical study with Alzheimer’s patients [64,65,66,67]. 

## 9. Prospects for GABA-R-Based Therapies in Clinical Applications

GABA seems well suited as a preventive or interventive therapy for new-onset T1D given its effectiveness in many studies of T1D rodents, its anti-inflammatory effects on human immune cells in vitro, and the results of a small clinical trial, as discussed above. Moreover, the combination of GABA treatment with other types of T1D therapeutic approaches such as antigen-specific immunotherapy or immunosuppressive agents yields enhanced beneficial effects over those of either monotherapy in diabetic mice [23,28,45,146,147,148,149,150]. Although not discussed herein, GABA treatment can also promote the proliferation of human and rodent insulin-producing ß-cells in diabetic animals (reviewed in [151]). 

Second, the ability of homotaurine to (1) inhibit neuroinflammation in animal models of MS and autism, and (2) reduce circulating levels of IL-18 in a small clinical trial with Alzheimer’s patients, together with its excellent safety profile in a phase III Alzheimer’s disease clinical trial, makes homotaurine a promising candidate to be an adjunctive treatment for MS. The advancement of GABA and homotaurine toward clinical trials for autoimmune diseases, however, faces challenges since these compounds are available over the counter in many countries, which limits pharmaceutical companies’ interest in their further development. 

It is widely opined that additional therapeutic approaches are urgently needed for the treatment of coronavirus infections because new SARS-CoV-2 variants are sure to arise, some of which may evade vaccine-induced immune responses or resist antiviral medications. Our observations in mice infected with two very different types of coronaviruses suggest that GABA treatment could be a generalizable off-the-shelf inexpensive treatment to help reduce the severity of illness caused by new SARS-CoV-2 variants and novel coronaviruses. The human equivalents of the doses of GABA and homotaurine that were used in the preclinical studies of autoimmune diseases and COVID-19 are within the levels already deemed to be safe for human use. Since GABA and homotaurine are stable at room temperature and inexpensive, they are promising candidates to help treat COVID-19, especially in developing countries. Perhaps GABA_A_-R ligands will augment the therapeutic efficacy of other treatments for COVID-19. We also speculate that homotaurine, or other BBB-permeable GABA_A_-R-specific ligands, may limit the neurological sequelae of COVID-19. It must be emphasized, however, that these contentions are largely based on results from preclinical models. The impact of GABA-R agonists on human immune cells, lung cells, and cells in the CNS differs in important ways from that of rodents. Indeed, the ability of GABA-R ligands to limit inflammatory immune responses could interfere with immune responses against SARS-CoV-2 and exacerbate COVID-19. Therefore, careful clinical trials are needed to ascertain whether, and at what stage of the disease process, treatment with a GABA-R agonist might be beneficial for COVID-19 patients. 

**Table 1 biomedicines-11-00254-t001:** Outstanding Questions.

How do GABA_A_-R and GABA_B_-R signaling pathways modulate intracellular signaling pathways and metabolome in different types of immune cells?
How does GABA act in paracrine and autocrine fashions on different immune cell functions?
Will tumor-reactive T cells that have been engineered to be GABA-R-deficient have a greater ability to destroy certain types of tumors?
What roles do GABA and GABA-Rs play in the progression of glioblastomas and neuroblastomas?
How do GABA_A_-R agonists reduce coronavirus load in the lungs?
Do GABA-R agonists modulate ACE2 expression, surfactant production/absorption, and/or inflammatory responses, and autophagy in human lungs?
How does GABA-R signaling modulate the innate immune responses of SARS-CoV-2 infected lung cells?
Can the anti-inflammatory effects of GABA_A_-R agonists on immune cells and CNS glial cells be exploited to reduce neuroinflammation and help ameliorate disorders such as MS and long COVID?

## Data Availability

Not applicable.

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
