# Peer review of "The GABA and GABA-Receptor System in Inflammation, Anti-Tumor Immune Responses, and COVID-19"

_biomedicines, 2023, doi:10.3390/biomedicines11020254_

Round 1
Reviewer 1 Report
The review article addresses the lesser known roles of GABA and GABA-Rs outside of the nervous system. The topic is introduced very logically by first summarizing the role of GABA in the CNS before examining roles for GABA and its receptors in various other contexts. The review appears to be very comprehensive in terms of citing relevant literature relevant to this topic. Overall it is a timely summary of a very interesting topic.
The review is likely to be of interest to a broad audience in areas including immunology, cancer biology and virology. A few improvements could be made to the manuscript to ensure that it is accessible to these various readers, who may not be familiar with GABA and its role in the CNS.
An introductory figure should be added in section 2 that illustrates the GABA-A and GABA-B receptors along with the basic pharmacology that is relevant in the remainder of the review. This could show that basics of GABA-B signalling in a neuronal context and also explain the factors that determine the direction of Cl- flux through GABA-A receptors.
The existing figure later in the manuscript could then focus on summarizing the GABA signalling in the various non-neuronal contexts. I suggest that this figure could be divided into separate panels for each context rather than trying to include all of these in one figure. As drawn, at first glance it looks like the figure is illustrating intracellular signalling inside one cell, whereas in reality, what is being described are the actions of GABA on different cell types and their interactions with each other. It would also be worth illustrating the source of endogenous GABA, in so far as that is known, in each context.
Section 1.
There are no citations to the literature in this section. The 3rd and 4th sentence in this section make quite specific points that should in my view be supported by relevant references.
Section 2.
In describing GABA-A receptors it is stated that they “let chloride passively flow in or out of the cell depending on the cell’s membrane polarity”. Also in section 7 it is stated that “The activation of GABAA-Rs leads to the in-flux or efflux of Cl-, depending on the cell membrane potential.”
My understanding is that it is not so much the cell membrane potential (or polarity) that determines the direction of Cl- flux, but rather the concentration gradient for Cl- in a particular context. A high extracellular Cl- concentration will result in influx and a high cyctosolic concentration will result in efflux when GABA-A channels open. The expression levels of chloride transporters determine this (in neurons at least and presumably also in other cells). A better explanation of this concept and the idea of Cl- reversal potential in cells would be beneficial - particularly for non-expert readers.
Section 3
Please provide a more precise description of the “autism-spectrum-like disorder in rats” that is referred to.
Section 6
Please spell out / explain the abbreviations ATII and ATI.
Section 7
Please spell out the abbreviations MHV-1 and SARS-CoV-2.
Author Response
We thank the Reviewers for their helpful comments, which we respond to point-by-point.
Reviewer #1
The review article addresses the lesser known roles of GABA and GABA-Rs outside of the nervous system. The topic is introduced very logically by first summarizing the role of GABA in the CNS before examining roles for GABA and its receptors in various other contexts. The review appears to be very comprehensive in terms of citing relevant literature relevant to this topic. Overall it is a timely summary of a very interesting topic.
The review is likely to be of interest to a broad audience in areas including immunology, cancer biology and virology. A few improvements could be made to the manuscript to ensure that it is accessible to these various readers, who may not be familiar with GABA and its role in the CNS.
An introductory figure should be added in section 2 that illustrates the GABA-A and GABA-B receptors along with the basic pharmacology that is relevant in the remainder of the review. This could show that basics of GABA-B signalling in a neuronal context and also explain the factors that determine the direction of Cl- flux through GABA-A receptors.
Response: We understand the Reviewer’s interest in the intracellular signaling mechanisms. We have focused this review on the effects of GABA-R agonists on immune cell populations and their potential use for disease management. We would like to only briefly refer to GABA-R-mediated intracellular changes within immune cells and tumor cells and provide the references for interested readers. We are not trying to encompass all aspects of the GABA system in the immune system. Also, while there is information on ion fluxes in neurons and immune cells soon after GABA-R activation, GABA is given quite long-term for the treatment of immune disorders and there are sure to be compensatory intracellular responses that then further modulate the intracellular ionic milieu. We think that the long-term changes in immune cell’s intracellular ionic milieu are complex and still terra incognita and so we want to paint a picture with only broad brush strokes.
The existing figure later in the manuscript could then focus on summarizing the GABA signalling in the various non-neuronal contexts. I suggest that this figure could be divided into separate panels for each context rather than trying to include all of these in one figure. As drawn, at first glance it looks like the figure is illustrating intracellular signalling inside one cell, whereas in reality, what is being described are the actions of GABA on different cell types and their interactions with each other. It would also be worth illustrating the source of endogenous GABA, in so far as that is known, in each context.
Response: We have modified the figure title and the figure itself to make it look less like intracellular signalling pathways. We have considered separating the different types of diseases into separate panels, but we feel that would lose sight of some of the interrelationships between various immune responses and disease states.
As far as illustrating the sources of endogenous GABA, that is a complex issue. For example, in tumors, GABA can come from B cells and from certain types of tumor cells. In the lungs, the cells that make GABA depends on whether looking in the upper or lower airways. I don’t think that we could generate a helpful figure and we think readers will be best served by referring to the references which provide granular histological information.
Section 1.
There are no citations to the literature in this section. The 3rd and 4th sentence in this section make quite specific points that should in my view be supported by relevant references.
Response: We left the 3rd and 4th sentences in the Introduction unreferenced because we have 50 or so references for those sentences since these sentences introduce a major focus of the review. Rather, we present the relevant references point-by-point within the text, which we think makes the material clearer to the reader. If the Reviewer would like, we can add “as discussed below”.
Section 2.
In describing GABA-A receptors it is stated that they “let chloride passively flow in or out of the cell depending on the cell’s membrane polarity”. Also in section 7 it is stated that “The activation of GABAA-Rs leads to the in-flux or efflux of Cl-, depending on the cell membrane potential.”
My understanding is that it is not so much the cell membrane potential (or polarity) that determines the direction of Cl- flux, but rather the concentration gradient for Cl- in a particular context. A high extracellular Cl- concentration will result in influx and a high cyctosolic concentration will result in efflux when GABA-A channels open. The expression levels of chloride transporters determine this (in neurons at least and presumably also in other cells). A better explanation of this concept and the idea of Cl- reversal potential in cells would be beneficial - particularly for non-expert readers.
Response: Thank you for pointing that out. We have amended the statements as suggested. The Introduction now states “GABAA-Rs and GABAB-Rs. GABAA-Rs are pentamers of 19 possible subunits that form fast-acting chloride channels, which when activated let chloride passively flow in or out of the cell. The direction of Cl- flux depends on the equilibrium potential of Cl- relative to the cell membrane potential, which in turn is modulated by the activity of the cell’s Cl- transporters [5, 6, 7]. The influx/efflux of Cl- through GABAA-Rs modulates the levels of intracellular ions and consequently many signaling pathways involved in neurodevelopment and neurotransmission [5, 6, 7].”
Similarly, section 7 now reads “The ability of GABA treatment to modestly reduce viral loads in both MHV-1 and SARS-CoV-2-infected mice was unexpected. The activation of GABAA-Rs leads to the influx or efflux of Cl- depending on equilibrium potential of Cl- relative to the cell membrane potential. Consequently, activating GABAA-Rs on adult neurons or islet a-cells leads to Cl- influx and hyperpolarization, while activation of GABAA-Rs on T cells, islet ß-cells, and lung ATII cells, causes Cl- efflux and depolarization [16, 17, 18, 101, 114, 133, 134]. Coronaviruses promote Ca2+ influx into host cells to enhance their replication [135, 136].”
Section 3
Please provide a more precise description of the “autism-spectrum-like disorder in rats” that is referred to.
Response: That sentence now reads: “In other studies of rats with an autism spectrum-like disorder caused by prenatal exposure to valproic acid [61], homotaurine treatment reduced the levels of IL-1ß, IL-6, and TNF-a, but increased the levels of IL-10 in the CNS, decreased neuronal loss, and ameliorated behavioral deficits [62].
Section 6
Please spell out / explain the abbreviations ATII and ATI.
Response: Thank you. We have now explained those abbreviations.
Section 7
Please spell out the abbreviations MHV-1 and SARS-CoV-2.
Response: We now spell out these abbreviates within Section 7.
Reviewer 2 Report
The review manuscript by Tian and Kaufam focuses on emerging experimental evidence that GABA and homotaurine, with a different ability to pass the BBB, can selectively activated GABA-receptor to act in a natural way as braking systems to limit inflammation in autoimmune diseases and in the CNS, respectively.
The beneficial effect of both moleculeson SARS-CoV2 infection and long-COVID is suggested based on recent studies, using rodents and therefore any translation to humans warrants further reserach.
The review ms is nicely written and is timely. It adds an additional interesting perspective on the use of these molecules in rather unexpected pathological conditions as summarized in Table 1.
I have only one request:
Page 6: the paragraph starting with "SARS-CoV-2 studies". At line 4 the authors say "We therefore studied transgenic......" until ".....COVID-19 (Fig. 1)" I do not understand to which study they have co-authored they refer to. they mention ref 124-130, but I cannot find the one with this specific results.
please clarify this important point and provide the correct reference
Author Response
Reviewer #2:
The review manuscript by Tian and Kaufman focuses on emerging experimental evidence that GABA and homotaurine, with a different ability to pass the BBB, can selectively activated GABA-receptor to act in a natural way as braking systems to limit inflammation in autoimmune diseases and in the CNS, respectively.
The beneficial effect of both molecules on SARS-CoV2 infection and long-COVID is suggested based on recent studies, using rodents and therefore any translation to humans warrants further research.
The review ms is nicely written and is timely. It adds an additional interesting perspective on the use of these molecules in rather unexpected pathological conditions as summarized in Table 1.
I have only one request:
Page 6: the paragraph starting with "SARS-CoV-2 studies". At line 4 the authors say "We therefore studied transgenic......" until ".....COVID-19 (Fig. 1)" I do not understand to which study they have co-authored they refer to. they mention ref 124-130, but I cannot find the one with this specific results. please clarify this important point and provide the correct reference.
Response: Thank you. We had neglected to cite our own recent work. We have now added that information.